# Single Nucleotide Polymorphisms in *MIR143* Contribute to Protection against Non-Hodgkin Lymphoma (NHL) in Caucasian Populations

**DOI:** 10.3390/genes10030185

**Published:** 2019-02-27

**Authors:** Gabrielle Bradshaw, Larisa M. Haupt, Eunise M. Aquino, Rodney A. Lea, Heidi G. Sutherland, Lyn R. Griffiths

**Affiliations:** Genomics Research Centre, School of Biomedical Sciences, Institute of Health and Biomedical Innovation, Queensland University of Technology, Brisbane, QLD 4001, Australia; gabrielle.bradshaw@hdr.qut.edu.au (G.B.); larisa.haupt@qut.edu.au (L.M.H.); eunise.aquino@connect.qut.edu.au (E.M.A.); rodney.a.lea@gmail.com (R.A.L.); heidi.sutherland@qut.edu.au (H.G.S.)

**Keywords:** biomarker, cancer, *MIR143*, miRSNP, non-Hodgkin lymphoma, rs17723799, Hexokinase 2, HKII

## Abstract

Recent studies show an association of microRNA (miRNA) polymorphisms (miRSNPs) in different cancer types, including non-Hodgkin lymphoma (NHL). The identification of miRSNPs that are associated with NHL susceptibility may provide biomarkers for early diagnosis and prognosis for patients who may not respond well to current treatment options, including the immunochemotherapy drug combination that includes rituximab, cyclophosphamide, doxorubicin, vincristine and prednisome (R-CHOP). We developed a panel of miRSNPs for genotyping while using multiplex PCR and chip-based mass spectrometry analysis in an Australian NHL case-control population (300 cases, 140 controls). Statistical association with NHL susceptibility was performed while using Chi-square (χ^2^) and logistic regression analysis. We identified three SNPs in *MIR143* that are to be significantly associated with reduced risk of NHL: rs3733846 (odds ratio (OR) [95% confidence interval (CI)] = 0.54 [0.33–0.86], *p* = 0.010), rs41291957 (OR [95% CI] = 0.61 [0.39–0.94], *p* = 0.024), and rs17723799 (OR [95% CI] = 0.43 [0.26–0.71], *p* = 0.0009). One SNP, rs17723799, remained significant after correction for multiple testing (*p* = 0.015). Subsequently, we investigated an association between the rs17723799 genotype and phenotype by measuring target gene Hexokinase 2 (*HKII*) expression in cancer cell lines and controls. Our study is the first to report a correlation between miRSNPs in *MIR143* and a reduced risk of NHL in Caucasians, and it is supported by significant SNPs in high linkage disequilibrium (LD) in a large European NHL genome wide association study (GWAS) meta-analysis.

## 1. Introduction

The global incidence of non-Hodgkin lymphoma (NHL) varies with the geographical region, with the highest incidence rates occurring in North America, Europe, and Australasia [1]. According to the American Cancer Society, non-Hodgkin lymphoma (NHL) is one of the most common cancers in the United States, with around 75,000 people being predicted to be diagnosed in 2018, resulting in around 20,000 deaths [2]. The European Cancer Information System predicts approximately 115,000 new NHL cases in 2018, resulting in 48,000 deaths [3]. Currently, in Australia, lymphoma is the sixth most commonly diagnosed cancer (most common is breast cancer, followed by prostate, colorectal, melanoma, and lung cancer) [4]. NHL is comprised of a large group of diverse lymphoid malignancies involving either the lymph nodes or extranodal sites and it is more prevalent in males living in more developed countries over the age of 60 years [5]. The two most common NHL subtypes are diffuse large B-cell lymphoma (DLBCL), which is highly aggressive with a poor prognosis [6] and follicular lymphoma (FL), which is more indolent, but it can undergo a high-grade transformation into DLBCL [7]. Molecular mechanisms that are involved in NHL pathogenicity have been widely investigated; however, there is still a need to identify improved biomarkers for clinical diagnostic and therapeutic use.

Mature microRNAs (miRNAs) are short sequences of non-coding RNA that function to regulate the translational expression of their target genes through complementary binding to the 3′-untranslated regions (3′-UTRs) of the target mRNA, in what is referred to as the miRNA recognition element (MRE) [8,9]. The processing pathway and biogenesis of miRNAs has been well characterised [10,11,12], with several miRNAs being involved in targeting and regulating the expression of oncogenes and/or tumour suppressor genes [13]. Single nucleotide polymorphisms (SNPs) in miRNA biogenesis genes and miRNA genes themselves influence miRNA biogenesis, the processing of precursors to mature miRNAs, regulatory function, and stability of the miRNAs, and, if located in the mRNA 3′-UTR binding sites, can also disrupt miRNA binding and affect the expression of these target genes [14]. According to the miRNA SNP Disease Database (MSDD) [15] SNPs that occur in miRNA-related functional regions, such as mature miRNAs, promoter regions, pri- and pre-miRNAs, and target gene 3′UTR binding sites are collectively referred to as “miRSNPs” (miRNA polymorphisms). These polymorphisms represent a novel category of functional molecules that regulate gene expression [16]. SNPs in miRNA genes have been shown to affect miRNA biogenesis, with these SNPs potentially resulting in reduced and increased mature miRNA levels. It has been predicted that, if a SNP occurs in the miRNA stem region and decreases the stability of the hairpin structure, then it will reduce the amount of mature miRNA produced [16]; however, if it fails to decrease stability, it will tend to increase the amount of mature miRNA produced and lead to disease pathogenesis [16]. Furthermore, SNPs occurring in the non-coding regulatory regions of genes, such as the miRNA-binding sites in the 3′-UTRs of target genes, can cause mRNA instability changes, leading to disease pathogenesis. A number of studies have identified roles for miRNAs [17], as well as miRSNPs [18,19] in NHL. These studies have examined the suitability of miRSNPs as diagnostic and prognostic biomarkers for improved clinical treatment and survival, particularly as NHL is a heterogeneous disease with outcomes that vary from patient to patient. However, few studies have demonstrated the association of SNPs in miRNAs with NHL risk. 

Serum lactate dehydrogenase A (LDH-A), which is an enzyme converting pyruvate to lactate, is commonly elevated in aggressive NHL cases, such as DLBCL and Burkitt’s lymphoma, and it has been used as a prognostic indicator for overall survival [5]. Cancer cells are able to produce lactate in a high oxygen environment (aerobic glycolysis), causing a phenomenon of cancer metabolism, known as the Warburg effect. The unusual utilisation of glucose by cancer cells results in the production of lactate, which is normally only produced during anaerobic respiration [20]. Hexokinase 2 (HKII) is a tissue-specific isoenzyme that phosphorylates glucose to glucose-6-phosphate (G-6-P) at the start of the glycolysis pathway, and its upregulation contributes to cancer cell glycolysis and the Warburg effect [21,22,23]. Hexokinase 2 maintains the high rate of glucose catabolism that is required for the survival of tumour cells, allowing them to sustain a higher rate of proliferation and resistance to cell death signals [21,23]. A variety of tumours are characterised by upregulated HKII expression, making it an attractive therapeutic target [23,24,25]. Oncogenes, such as *MYC* and *RAS*, and tumour suppressor genes, such as *p53*, are known to regulate metabolic enzyme expression and they can cause cancer if they happen to undergo mutations [20]. In addition, hypoxia-inducible factor 1-alpha (HIF1A) and MYC proteins cooperate to regulate the expression of *HKII* and *PDK1* genes [26]. An investigation of the Warburg effect in Burkitt’s lymphoma (BL) cells, where the HIF1A protein was highly expressed in EBV-positive BL cell lines, showed that the inhibition of MYC activity led to decreased expression of MYC-dependent genes and LDH-A activity, implicating MYC as the master regulator of aerobic glycolysis in these cells [26]. 

Most recently, Bhalla et al. investigated the role of hypoxia in DLBCL [27]. They demonstrated that the up-regulation of HIF1A resulted in repressed protein translation, however HKII was selectively translated by eIF4E1 to promote DLBCL growth in vitro and in vivo under hypoxic stress. Their findings suggest HKII as a key metabolic driver of the DLBCL phenotype. It has also been shown that acquired resistance in rituximab-resistant lymphoma cell lines (RRCL) was associated with the deregulation of glucose metabolism and an increase in the apoptotic threshold, leading to chemotherapy resistance, where RRCL expressed higher levels of HKII. Targeting HKII in these cells led to decreased resistance, implying that increased HKII levels in aggressive lymphoma causes chemotherapy resistance, while also identifying this as a potential therapeutic target [22]. Many HKII inhibitors have been effective in anti-cancer therapies, such as 3-bromopyruvate (3-BP), which was found to inhibit HKII, activate the mitochondrial cell death pathway, and deplete levels of ATP [22], and it was also shown to induce apoptosis in a breast cancer cell line (MDA-MB-231) [28]. 

The main aim of this study was to investigate the genetic association between miRSNPs that were previously implicated in tumourigenesis and/or NHL susceptibility and prognosis, based on a comprehensive review of recent literature [19,29,30]. We genotyped 39 miRSNPs using a multiplex PCR and matrix assisted laser desorption time-of-flight (MALDI-TOF) mass spectrometry (MS) MassARRAY^®^ system in our Genomics Research Centre Genomics Lymphoma Population (GRC GLP-non-Hodgkin lymphoma) cohort. After basic association testing, three SNPs in *MIR143* were identified to be significantly associated with NHL, with one SNP, rs17723799, remaining significant after Bonferroni correction for multiple testing (*p*-value = 0.015). After logistic regression testing the same three SNPs in *MIR143* were significantly associated with reduced risk of NHL in the Additive model: rs3733846 (Odds ratio (OR) [95% confidence interval (CI)] = 0.54 [0.33–0.86], *p* = 0.010), rs41291957 (OR [95% CI] = 0.61 [0.39–0.94], *p* = 0.024), and rs17723799 (OR [95% CI] = 0.43 [0.26–0.71], *p* = 0.0009). As *HKII* is a known target gene for mature hsa-miR-143 (miR-143), our secondary aim examined HKII expression in four patient-derived NHL cell lines, as compared to a metastatic breast cancer (MDA-MB-231) and melanoma (MDA-MB-435) cell line, as well as two healthy control subjects to assess the potential functional link between miR-143 regulation and HKII levels in NHL.

## 2. Materials and Methods 

### 2.1. Study Population 

The GRC-GLP retrospective cohort consists of 300 NHL cases and 140 healthy controls. All of the samples are of Caucasian origin with Australian/British/European grandparents with no family history of a haematological malignancy. The cases were matched according to age- (within five years), sex-, and ethnicity with healthy cancer-free controls. Cases were collected between 2010 and 2014 from the Princess Alexandra Hospital in Brisbane, and the GRC clinic in Mermaid Waters on the Gold Coast. The case cohort mainly consists of FL (*n* = 95) and DLBCL (*n* = 88), with 79 cases being unclassified as NHL or “Other B-cell”. B-cell chronic lymphocytic lymphoma (CLL), cutaneous T-cell lymphoma, Mantle cell lymphoma (MCL), Splenic marginal zone lymphoma (SMZL), Mucosa-associated lymphoid tissue lymphoma (MALT), and Burkitt’s lymphoma (BL) make up the remaining subtypes, in the order of frequency from highest to lowest (Table 1). Patients and healthy volunteers were required to complete a personal questionnaire and provide written consent to participate in research. The cohort is comprised of 48% male and 52% female participants, with the mean age of cases 63.72 years (standard deviation (SD) = 12.95 years) and the mean age of controls 63.14 years (SD = 13.03 years). In addition, 35 new NHL cases were received in 2016 (collected in 2014), comprised of 71% male and 29% female, with an average age of 59.6 years (SD = 13.6 years). Ethics for the collection and the use of participant samples was approved by the Queensland University of Technology Research Ethics Committee (approval number 1400000125). All of the subjects gave written informed consent, in accordance with the Declaration of Helsinki.

### 2.2. Genomic DNA Extraction 

Genomic DNA (gDNA) was extracted from whole blood collected into EDTA tubes using an in-house salting-out method, as evaluated by Chacon et al. [31]. DNA concentration and purity was measured using the NanoDrop^TM^ ND-1000 spectrophotometer (ThermoFisher Scientific Inc., Waltham, MA, USA) before dilution to 15–20 ng/µL and storage as stock gDNA at 4 °C. 

### 2.3. miRSNP Selection and iPlex Primer Design

39 cancer-related microRNA-related SNPs (miRSNPs) were selected for genotyping by chip-based multiplex PCR and MALDI-TOF MS (Agena MassARRAY^®^, San Diego, CA, USA), following a comprehensive review of the recent literature related to miRNA biomarkers in tumourigenesis, including breast cancer and/or non-Hodgkin lymphoma [18]. Forward, reverse, and extension primers were designed for each SNP using the MassARRAY^®^ AgenaCx design software. Primers were manufactured by Integrated DNA Technologies (IDT, Singapore) and primer sequences are available on request. 

### 2.4. Primary Multiplex PCR

Genotyping was performed using the iPlex^TM^ GOLD Reagent Kit (Agena), according to the manufacturer’s instructions. Forward and reverse primers were pooled, with extension primers being pooled according to mass using the linear primer adjustment method. All of the PCR reactions were performed in 96-well reaction plates in a Veriti^TM^ 96-well Thermal Cycler (ThermoFisher Scientific Inc., Waltham, MA, USA).

### 2.5. MALDI-TOF MS and Data Analysis

Prior to each run, the extension products were spotted on to the SpectroCHIP^®^ (Agena) using the Agena MassARRAY^®^ Nanodispenser (Agena), which was immediately loaded into the Agena MassARRAY^®^ Analyser 4 for genotype detection of each SNP in the assay by the SpectroAcquire v4.0 software (Agena). Final data analysis was performed using the MassARRAY^®^ Typer software v4.0 (Agena). 

### 2.6. In-Vitro Culture of Cell Lines and Primary Lymphocytes

All of the commercial patient-derived NHL B-lymphoid (SU-DHL-4, Raji, Mino, Toledo), breast cancer (MDA-MB-231), and melanoma (MDA-MB-435) cell lines were previously acquired from the American Type Culture Collection (ATCC^®^, Manassas, VA, USA). The cell lines were authenticated by the GenePrint^®^ 10 System, according to the manufacturer’s instructions as a service provided by the School of Biomedical Sciences, QUT. For experiments, the cells were seeded into T-75 flasks with 20 mL RPMI-1640 or DMEM growth media (Invitrogen, ThermoFisher), supplemented with 10% foetal bovine serum and 1% Penicillin-Streptomycin antibiotic. Cultures were incubated at 37 °C under 5% CO_2_ conditions. The cells were harvested into TRIzol^®^ and Runx protein-lysis buffer for RNA and protein extraction (see below) and then pelleted for DNA extraction at 90% confluence and 95% viability. Cell counts and viability were assessed via the Trypan Blue exclusion method on a TC10^TM^ automated cell counter (Bio-Rad, Hercules, CA, USA). Primary lymphocytes from healthy subjects were isolated from peripheral blood mononuclear cells (PBMCs) using the Ficoll-Histopaque centrifugation method (Sigma-Aldrich, St. Louis, MI, USA). Cells were washed with 1x PBS and then plated into T-75 culture flasks in RPMI-1640 with Phytohaemagglutinin (PHA) (Sigma-Aldrich) mitogen for up to 24 h to allow for monocytes to adhere to the flask. After a minimum of three hours, the lymphocytes in suspension were removed and plated into new flasks with PHA and interleukin 2 (IL-2) (Sigma-Aldrich). The lymphocytes were left to proliferate at 37 °C under 5% CO_2_ conditions for five days. Live lymphocyte counts were performed on a haemocytometer and viability assessed by Trypan blue dye. 

### 2.7. Validation of Genotyping by MassARRAY^®^ and Genotyping of Cell Lines and Healthy Controls by Sanger Sequencing

The genotype plots on the MassARRAY^®^ system were assessed and manually genotyped for scattered data not assigned by the analysis software. For statistically significant SNPs (*p* < 0.05), a subset of samples was validated by Sanger sequencing (SS) using primers (Table 2) to ensure the accuracy of genotype calls by the Agena MassARRAY^®^ system. Briefly, amplified PCR products were treated with Exo-SAP for use in Big Dye Terminator v3.1 (ThermoFisher Scientific, Life Technologies) sequencing reactions, and analysed on the 3500 Genetic Analyser (ThermoFisher Scientific, Life Technologies). Sequencing data for each sample chromatogram was assessed using Chromas Lite 2.1.1 software). The genotypes obtained were found to be 100% concordant between the MassArray^®^ and SS. gDNA was extracted from NHL cell line pellets using the GenePrint^®^ 10 System, according to the manufacturer’s instructions. 

### 2.8. RNA Extraction, cDNA Synthesis and q-PCR

Total RNA was extracted from cell lines and control primary lymphocyte homogenates using the Direct-zol^TM^ RNA MiniPrep extraction kit (Zymo Research, Irvine, CA, USA), according to the manufacturer’s instructions. Total RNA was converted to complementary DNA (cDNA) while using the iScript^TM^ cDNA Synthesis Kit (Bio-Rad Laboratories), according to the manufacturer’s instructions. Each q-PCR was performed in triplicate, with 120 ng cDNA being amplified with SYBR^®^ Green PCR master mix (Promega, Madison, Wisconsin, USA) and *HKII* forward (5′-CCCGGGAAAGCAACTGTTTG-3′) and reverse (5′-ACCGGTGTTGAGAAGCTCTG-3′) primers in a 10ul final reaction volume on the QuantStudio 7 instrument under the following reaction conditions: 50 °C for 2 min (×1 cycle), 95 °C for 3 min (×1 cycle), 95 °C for 3 s, and 60 °C for 30 s (×50 cycles). PCR efficiency for the *HKII* primer pair was 103%, with a single peak melt curve. *18S* was used as an endogenous control and gene expression was calculated using the relative quantification method (^ΔΔ^Ct). A student’s *t*-test was used to calculate the significant differences between *HKII* expression profiles in cell lines and healthy controls. 

### 2.9. Protein Detection by Western Blot

Total protein was isolated from cell lysates using the RUNX protein-lysis buffer (20 mM HEPES, 25% glycerol, 1.5 mM MgCl_2_, 420 mM NaCl, 0.5 mM DTT, 0.2 mM EDTA, 0.5% Igepal CA-630, 0.2 mM Na_3_VO_4_, 1 mM PMSF, and dH_2_0 containing a protease and phosphatase inhibitor cocktail). Protein concentration was measured using the Qubit^TM^ Protein Assay Kit (Invitrogen). For SDS-PAGE (Sodium Dodecyl Sulphate-Polyacrylamide Gel Electrophoresis), 30 µg total protein was separated for 1.5 h at a constant 120 V in a mini-PROTEAN Tetra unit. The HKII protein (109 kDa) was detected using a mouse anti-HKII primary antibody (Abcam, ab104836) and an HRP-conjugated anti-mouse IgG secondary antibody (Cell Signaling, #7076). Sample loading was normalised using secondary HRP-conjugated anti-Beta-actin (Cell Signaling, #5125S). Enhanced chemiluminescence detection (Clarity^TM^, ECL, Bio-Rad) of HKII and Beta-actin was performed using the Fusion Spectra chemiluminescent system (Vilber Lourmat, Fisher Biotec) and optical density quantitation was assessed using Image J software [32].

### 2.10. Statistical Analysis

Chi-square (χ^2^) analysis was used to determine the differences in genotype and allele frequencies between case and control samples, where a *p*-value < 0.05 was considered to be statistically significant. Genetic association and haplotype analysis of genotyping data was performed using the latest Plink v1.09 [33], RStudio [34], and HaploView [35] software. Quality control (QC) filters were applied to the full dataset consisting of 39 variants, 300 cases, and 140 controls. Following basic association testing, where correction for multiple testing was applied in the filtered dataset, a mixed model logistic regression test adjusting for the covariate ‘sex’ was performed in association with NHL with risk determined by OR and CI that were set at 95%. 

### 2.11. NHL-GWAS Replication Dataset

Cerhan et al. [36] conducted a meta-analysis of three new NHL genome-wide association studies (GWAS): NCI, GELA, and MAYO_DLBCL, and one previous scan (SF), including around 3857 cases and 7665 healthy controls of European ancestry (InterLymph Consortium), to identify the genetic susceptibility loci for DLBCL, with further genotyping of nine SNPs in 1359 cases and 4557 healthy controls. For the first three groups, NCI, GELA, and MAYO_DLBCL, data was genotyped and for the scan, SF, data was imputed. We requested genotype summary statistics from the authors, for SNPs to be included in the GWAS that were in high linkage disequilibrium (LD) (D’ = 1.00), with significantly associated SNPs in our SNP MassARRAY^®^ panel, i.e., rs12659504 and rs878008, in order to investigate the replication of our findings. 

## 3. Results

### 3.1. Genetic Association of MIR143 SNPs and NHL Risk

In this study, 39 cancer-related microRNA-related SNPs (miRSNPs) were selected for genotyping by chip-based multiplex PCR and MALDI-TOF MS in the GLP-NHL population of 300 cases and 140 healthy controls. Allele frequencies for each variant in the GLP-NHL case-control cohort were established and the minor allele frequencies (MAFs) were compared to those that were observed in European and British populations in the Hapmap 1000G database (Table 3). After the QC filters were applied, basic allelic association testing revealed two SNPs in *MIR143/145* (rs3733846 [OR = 0.56, *p* = 0.012], rs41291957 [OR = 0.56, *p* = 0.008]), and one SNP in the promoter region of *MIR143* (rs17723799 [OR = 0.42, *p* = 0.0004]) on chromosome 5 to be significantly associated with a reduced risk of NHL (Table 3). After correction for multiple testing, the rs17723799 SNP remained statistically significant using the Bonferroni (*p* = 0.015) adjustment method (Table 3). A logistic regression test adjusted for the covariate ‘sex’ revealed the same three SNPs in *MIR143* to be significantly associated with the reduced risk of NHL in the Additive model: rs3733846 (OR [95% CI] = 0.54 [0.33–0.86], *p* = 0.010), rs41291957 (OR [95% CI] = 0.61 [0.39–0.94], *p* = 0.024), and rs17723799 (OR [95% CI] = 0.43 [0.26–0.71], *p* = 0.0009) (Table 4). Genotype and allele frequencies were determined for the three *MIR143* SNPs and were compared with 1000G and gnomAD MAF percentages (Table 5). The rs17723799 SNP was identified to have a significant difference in allele (*p* = 0.013) and genotype (*p* = 0.039) frequencies between the case and control sample cohorts (Table 5). A mixed model logistic regression test adjusted for the covariate ‘sex’ revealed the rs17723799 SNP to be significantly associated with a reduced risk of NHL in the Additive (OR [95% CI] = 0.43 [0.26–0.71], *p* = 0.0009), Dominant (OR [95% CI] = 0.54 [0.33–0.88], *p* = 0.015), Over-dominant (OR [95% CI] = 0.56 [0.33–0.92], *p* = 0.024), and Log-additive (OR [95% CI] = 0.58 [0.38–0.90], *p* = 0.017) models of inheritance (Table 6). The two other *MIR143* SNPs in LD with rs17723799, i.e., rs3733846, and rs41291957 (Table 7) were also shown to be associated with a significantly reduced risk of NHL after logistic regression testing in the Additive model only: rs3733846 (OR [95% CI] = 0.54 [0.33–0.86], *p* = 0.010) and rs41291957 (OR [95% CI] = 0.61 [0.39–0.94], *p* = 0.024). Analysis of the three SNPs was also performed in association with the two most common NHL subtypes that are present in the cohort, i.e., DLBCL and FL, however the results were non-significant for all three SNPs (*p* > 0.05). This could be due to the small number of cases representing these subtypes and the reduced statistical power. All three SNPs were validated by Sanger sequencing in 5–10 randomly selected cases and controls, with the genotype of all samples being confirmed. 

### 3.2. Replication of Summary Statistics in a Large EUROPEAN NHL GWAS Meta-Analysis

As we did not have access to an in-house replication NHL population, we interrogated the complete list of SNPs (700,000+ loci) that are available on the HumanOmniExpress-12-v1-1-C chip (Illumina) studied in three NHL GWAS for the presence of SNPs found to be significantly associated with NHL in our study. Unfortunately, our three SNPs in *MIR143* were not included in the GWAS, however we were able to find two SNPs (rs12659504 and rs878008) on chromosome 5 in the GWAS in high LD (r^2^ = 1.00, D’ = 1.00) with rs3733846 and rs17723799 (Table 7). A high linkage disequilibrium (LD) score identified strong non-random association of nearby variants on the same chromosome. Additionally, a meta-analysis of three individual GWAS and one imputation scan performed by Cerhan et al. [37], which included 3855 cases and 7664 controls, showed a significantly reduced risk of NHL for both of these SNPs: rs12659504 (OR [95% CI] = 0.91 [0.84–0.99], *p* = 0.033) and rs878008 (OR [95% CI] = 0.92 [0.84–1.00], and *p* = 0.041 (Table 8) (InterLymph Consortium).

### 3.3. Haplotype Analysis of MIR143 SNPs in LD on Chromosome 5

We performed haplotype analysis for the three SNPs (rs3733846, rs41291957, and rs17723799) that were located in *MIR143* on chromosome 5 found to be significantly associated with a reduced risk of NHL in our case-control cohort. The most commonly inherited haplotype was haplotype 3 (A-G-C), with a frequency of 0.82 (82%) in case samples and 0.79 (79%) in control samples (Table 9). A significantly reduced risk of NHL was observed with the inheritance of haplotype 4 (G-A-T) with all three minor alleles, which occurred at a frequency of 0.09 (9%): OR [95% CI] = 0.42 [0.18–1.00], *p* = 0.049) (Table 10). A linkage disequilibrium plot showed moderate linkage between the three SNPs when being assessed using Haploview 4.2 (Figure 1). 

### 3.4. Genotyping of rs17723799 (C>T) in Cell Lines and Healthy Control Samples

We genotyped four commercially sourced, immortalised NHL cell lines (SU-DHL-4, Raji, Mino and Toledo), a metastatic breast cancer cell line (MDA-MB-231), a melanoma cell line (MDA-MB-435), and two healthy cancer-free control subjects for the *MIR143* rs17723799 SNP. All of the NHL cell lines were homozygous for the wild type risk allele C (CC). The breast cancer cell line was homozygous wild type CC. The melanoma cell line was homozygous for the minor allele T (TT). Both the healthy control subjects were homozygous for the wild type C allele (CC). As the minor T allele is reported to be present at a frequency of 0.11 (11%) in the healthy European population (Table 5), these results were not surprising; however, it was interesting that all NHL cell lines were identified to be homozygous for the wild type allele C, which was observed as the risk allele for NHL in our genetic association analysis. 

### 3.5. miR-143 Target Gene HKII Expression in Cancer Cell Lines and Healthy Control Lymphocytes

miR-143 inhibits the expression of its target gene *HKII* by binding to a conserved recognition motif that is located in the 3′-UTR of mRNA. We examined *HKII* expression at the mRNA and protein level in four NHL cell lines (SU-DHL-4, Raji, Mino, Toledo), a metastatic breast cancer cell line (MDA-MB-231), and a melanoma cell line (MDA-MB-435). In addition, two healthy control donors were included as a reference for basal *HKII* expression levels for statistical analysis. 

#### 3.5.1. HKII Gene Expression Is Increased in NHL Compared to Healthy Controls

q-PCR analysis of *HKII* mRNA transcript expression showed increased levels of *HKII* mRNA transcript in the four patient-derived NHL cell lines (SU-DHL-4, Raji, Mino, and Toledo) when compared to lymphocytes that are derived from two healthy control subjects, however the differences in expression levels were not statistically significant (Figure 2). Western blot and band quantitation analysis confirmed these findings, showing increased HKII protein levels in the NHL cell lines examined when compared to the controls (Figure 3). 

#### 3.5.2. Increased HKII Gene Expression May Be Associated with the MIR143 rs17723799 TT Genotype

q-PCR analysis demonstrated *HKII* mRNA transcript levels that were were higher in the melanoma (MDA-MB-435) cells, correlating with the observed homozygous mutant rs17723799 TT genotype in these cells. *HKII* mRNA transcript levels were observed to be lower in all four NHL cell lines, and even lower in the breast cancer (MDA-MB-231) and healthy control cells, which correlated with the homozygous wild type rs17723799 CC genotype of these cells (Figure 2). A Student’s *t*-test showed a significant difference in the relative *HKII* mRNA expression between melanoma cells and the breast cancer (*p* = 0.044), control subject 1 (*p* = 0.045), and control subject 2 (*p* = 0.037) cells. Western blot and band quantitation analysis demonstrated upregulated HKII protein expression in the melanoma cells, with lower expression being observed in the breast cancer cells and even lower in the four NHL cell lines examined (Figure 3). Surprisingly, the breast cancer cells expressed low *HKII* mRNA transcript, but high protein levels. 

## 4. Discussion

Our study investigated 39 miRSNPs previously implicated in human tumourigenesis that were identified from the literature, including those not previously reported to have an association with NHL susceptibility, but rather with disease prognosis, in Caucasians. Following genotyping by multiplex PCR and MALDI-TOF MS, subsequent genotypic association and haplotype analysis identified three SNPs on chromosome 5 (rs3433846, rs17723799, and rs41291957) in *MIR143* in moderately high LD, which together appear to confer protection against NHL in our GRC-GLP cohort. In particular, the rs17723799 SNP in the promoter region of *MIR143* remained significant at the *p* < 0.05 threshold after Bonferroni correction for multiple testing. We also provided evidence for replication in a large European NHL GWAS meta-analysis study.

One study identified the rs11614913 SNP (T > C) in hsa-miR-169-a2 (miR-196-a2) to be associated with NHL in a Chinese case-control cohort, where the presence of the risk C allele and CC genotype was shown to be more frequent in the case samples, conferring an increased risk (OR [95% CI] = 1.82 [1.16–2.85], *p* = 0.009) [36]. The authors confirmed that the variant genotype affected the expression of miR-196-a2, where the carriers of the CC genotype had significantly higher levels than those with only one C allele or the TT genotype. Interestingly, this SNP was also shown to increase risk of hepatocellular carcinoma in a Chinese cohort of 109 cases and 105 controls [38]; however, no functional analyses were performed. The rs11614913 SNP was shown to be associated with decreased central nervous system (CNS) Acquired Immunodeficiency Syndrome (AIDS)-NHL in an American case-control cohort of 180 cases and 529 controls, where the CT genotype was more frequently observed in control samples and conferred protection (OR [95% CI] = 0.52 [0.27–0.99]. This study also showed an increased risk of systemic AIDS-NHL with the presence of the T allele in rs2057482, which creates a binding site for miR-196-a2 in the *HIF1A* 3′-UTR (OR [95% CI] = 1.73 [1.12–2.67]) [39]. We were not able to show a positive significant association for miR-196-a2 rs11614913 in our Australian/European cohort (OR [95% CI] = 0.93 [0.67–1.29], *p* = 0.668) (Table 3). This is most likely due to ethnicity differences in our cohort and the Chinese cohort of Li et al. [36]. Previous studies on this SNP have shown a significant association with cancer in predominantly Asian populations, but not in Caucasians, which may explain the lack of association in our cohort [40]. 

Other studies have shown the association of *MIR143/145* SNPs with reduced cancer risk in Chinese cohorts. Li et al. [41] conducted a case-control analysis of 12 SNPs in the promoter region of *MIR143/145* in 242 cases with colorectal cancer (CRC) and 283 healthy controls. In support of our findings in NHL, the rs3733846 (A > G) mutant genotype A/G was associated with a significantly reduced risk of CRC (OR [95% CI] = 0.57 [0.44–0.73], *p* < 0.001). A more recent study by Wu et al. [42] showed a functional association between the rs353293 SNP (G>A) AG/AA genotypes and a reduced risk of bladder cancer in 333 Chinese cases when compared to 536 healthy controls (OR [95% CI] = 0.64 [0.46–0.90], *p* = 0.008). In their study, in vitro luciferase reporter analysis was used to show a significantly reduced effect of the protective rs353293A allele as compared with the rs353293G allele on transcriptional activity (*p* < 0.001). The promoter transcriptional activity was identified to be reduced due to the polymorphism, which could also be the case for the rs17723799 SNP that we identified, however this would imply lower miR-143/145 levels and lower tumour suppressor activity, which appears to be in contradiction with their genotypic analysis data as well as ours, showing the protective effect of the polymorphism. The authors were not able to show whether serum miR-143/145 or the target gene expression levels were altered.

Epigenetic mechanisms, such as miRNAs, have been shown to regulate aerobic glycolysis in cancer cells through the targeting and down-regulation of glycolytic enzymes [14]. miR-143, an essential regulator of glycolysis [43], inhibits *HKII* expression by binding to a conserved recognition motif that is located in the 3′-UTR of *HKII* mRNA and has been observed to be inversely correlated with HKII levels in primary keratinocytes and in head and neck squamous cell carcinoma (HNSCC)-derived cell lines [44]. miR-143 and miR-155 have both been shown to regulate glycolysis by targeting *HKII* in breast cancer, with miR-155 being able to suppress miR-143 production through targeting the *C/EBP*β transcription activator for miR-143 [45]. In addition, the transcriptional regulation of miR-145 was recently reviewed by Zeinali et al. [46]. The authors discuss miR-145 regulation by DNA-binding factors, including c-MYC, p53, forkhead transcription factors of the O class 1 and 3 (Fox01 and Fox03), and miR-143/145 level regulation by RREB1 in other cancers but not in lymphoma.

The significant *MIR143* SNP that was identified in this study (rs17723799) has been previously reported via in-silico analysis to affect aerobic glycolysis. The authors showed that this SNP overlaps with the transcription factor binding site in the *MIR143* promoter region (148783674-148788779, UCSC Genome Browser) and concluded that this SNP could potentially affect the regulation of miRNA biogenesis and alter the hsa-miR-143-3p and hsa-miR-143-5p levels, thereby affecting its target genes that directly or indirectly control glycolysis [14].

Following our genotypic analysis, *HKII* expression at the mRNA and protein level in NHL (SU-DHL-4, Raji, Mino, Toledo), breast cancer (MDA-MB-231), and melanoma (MDA-MB-435) cell lines was assessed to determine the potential functional link between *MIR143* regulation by a promoter polymorphism and *HKII* target gene expression in these cells, along with lymphocytes from two healthy control subjects. Our results showed higher *HKII* mRNA and HKII protein levels in the NHL cell lines when compared to healthy controls, however the levels were lower in the NHL cell lines compared to the breast cancer (protein only) and melanoma cell lines. Melanoma cells carrying the *MIR143* rs17723799 homozygous recessive TT genotype expressed a significantly increased level of *HKII* mRNA transcript and HKII protein in comparison to other cancer cells that were carrying the rs17723799 homozygous wild type CC genotype, indicating that this homozygous polymorphism may have an effect on the transcriptional regulation of *MIR143*. Interrogation of the GeneCards Database identifed 38 promoters and enhancers for the *MIR143* gene with numerous transcription factor binding sites for each, implicating multiple mechanisms of regulation of this gene by multiple transcription factors in different tissue types. As we do not have access to an appropriate wild type melanoma cell line control, further studies on *MIR143* SNPs in melanoma may be beneficial. Our results also show an up-regulation of the HKII protein in the metastatic breast cancer (MDA-MB-231) cells, as reported by Palmieri et al. [24]. Interestingly, Western blot showed HKII levels were higher in the Toledo (DLBCL) cell line when compared to the other less aggressive NHL cell line subtypes. 

Although not in NHL, but as a marker of non-small cell lung cancer (NSCLC), miR-143 expression in peripheral blood mononuclear cells (lymphocytes and monocytes) was significantly lower in NSCLC patients than in healthy individuals (*p* < 0.0001) [47]. Similarly, a study in renal cell carcinoma (RCC) identified the miR-143/145 cluster to be downregulated in RCC tissues when compared to adjacent non-cancerous tissues, with significantly higher HKII levels in RCC tissues as compared to non-cancerous tissues, confirming the tumour suppressive effect of the miR-143/145 cluster through targeting HKII [48]. One study in aggressive Burkitt’s lymphoma (BL) showed that upregulated miR-143 by the regulation of PI3K/Akt prevented cell growth, implicating miR-143 as a tumour suppressor in NHL [49]. miR-143 has also been demonstrated to have an anti-tumour effect in leukaemia patients, where significantly lower miR-143 levels were observed in patients when compared to healthy controls, and the overexpression of miR-143 decreased DNA methyltransferase 3A (*DNMT3A)* mRNA and protein expression, thereby reducing cell proliferation, colony formation, and cell cycle progression, as well as increased apoptosis [50]. As there is a paucity of literature on *MIR143* and miR-143 expression and polymorphisms in NHL and its subtypes, we searched for significant eQTLs (expression quantitative trait loci) for the three analysed *MIR143* SNPs in the GTEx Portal [51]. GTEx analysis indicated increased *MIR143* host gene expression in skeletal muscle for heterozygous and homozygous mutant alleles in all three significant *MIR143* SNPs analysed (Appendix A). Although the relationship between miR-143/145 and its *MIR143* host gene is an open issue, the potentially increased *MIR143* host gene expression, as seen due to the polymorphisms in skeletal muscle, may enhance hsa-miR-143-3p and -5p processing, which poses a potential protective function for these SNPs in lymphocytes. The presence of the rs17723799 CT or TT genotype may increase *MIR143* expression and miR-143 transcription with the subsequent downregulation of *HKII* causing reduced glycolysis and lymphomagenesis. Although we did not measure mature miR-143 levels in the NHL cell lines or patient samples to correlate this with HKII expression, our genotypic analysis data indicate a possible protective functional effect of *MIR143* SNPs on miR-143 regulation, with a possible inhibitory effect on *HKII* expression in controls as compared to NHL cases. Further studies investigating *MIR143* and hsa-miR-143-3p/5p expression levels in different NHL subtypes and their association with HKII levels are therefore necessary. We have summarised the role of HKII and miR-143 in cancer in Figure 4a and a potential functional effect of the rs17723799 polymorphism in control cells when compared to NHL cells in Figure 4b.

A limitation of this study is the lack of expression data for mature hsa-miR-143-3p and hsa-miR-143-5p in the NHL cell lines or patient samples. We only had access to four NHL cell lines and we did not have access to RNA or protein from our retrospective patient cohort. Additionally, we were not able to compare HKII expression in NHL cell lines polymorphic for the rs17723799 SNP, as they were all wild type CC, and therefore further studies investigating miR-143 and HKII expression in NHL cell lines and/or patient-derived tumour samples with the CT or TT genotypes would be beneficial. 

## 5. Conclusions

This study is the first to report significant statistical correlation between SNPs in *MIR143* and reduced risk of NHL in Caucasians, and it is supported by the identification of significant SNPs in high LD in a large European NHL GWAS meta-analysis study. Our findings suggest that the three SNPs in LD in *MIR143* may be novel useful biomarkers to assess the risk of NHL in a clinical setting. Furthermore, this study gives some insight into *MIR143* transcriptional regulation by a protective promoter polymorphism in NHL. We do not suggest that there is a definitive functional association between *MIR143* SNPs and NHL onset, however further functional analyses are necessary to confirm this potential mechanism.

## Figures and Tables

**Figure 1 genes-10-00185-f001:**
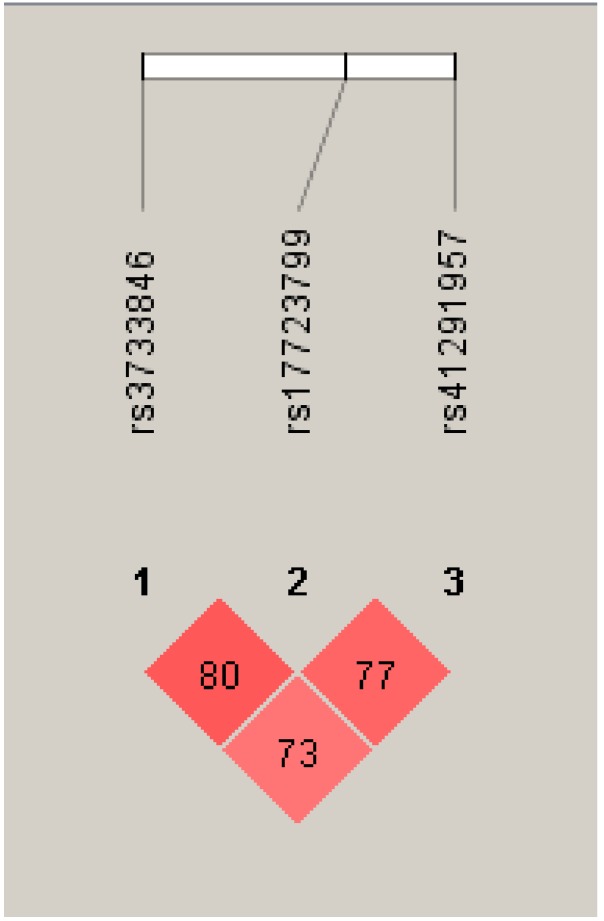
Linkage disequilibrium (LD) plot for analysed polymorphisms in the *MIR143* host gene using Haploview software version 4.2 (Daly Lab, Broad Institute, Cambridge, MA, USA).

**Figure 2 genes-10-00185-f002:**
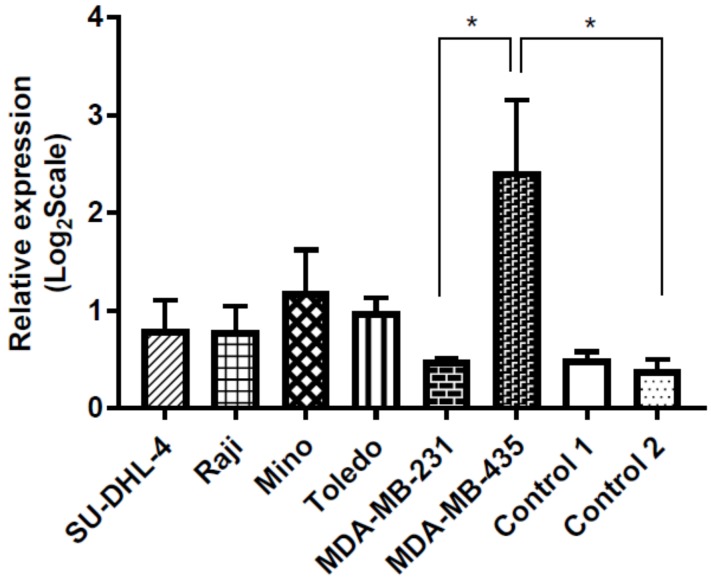
**Hexokinase 2 (*HKII*) gene expression**. q-PCR analysis of *HKII* mRNA transcript expression in NHL (SU-DHL-4, Raji, Mino, Toledo), breast cancer (MDA-MB-231) and melanoma (MDA-MB-435) cell lines as compared to healthy controls 1 and 2. *HKII* mRNA levels were increased in NHL cell lines compared to breast cancer cells and healthy controls, however expression levels were not significantly different. *HKII* mRNA levels were significantly increased in the melanoma (MDA-MB-435) cells when compared to breast cancer cells (MDA-MB-231) (*p* = 0.044), healthy control 1 (*p* = 0.045) and 2 (*p* = 0.037), but not significantly increased compared to the NHL cells. Relative expression normalized to *18S*, error bars = SEM and statistical significance calculated using paired Student’s *t*-test and defined as: * *p* < 0.05.

**Figure 3 genes-10-00185-f003:**
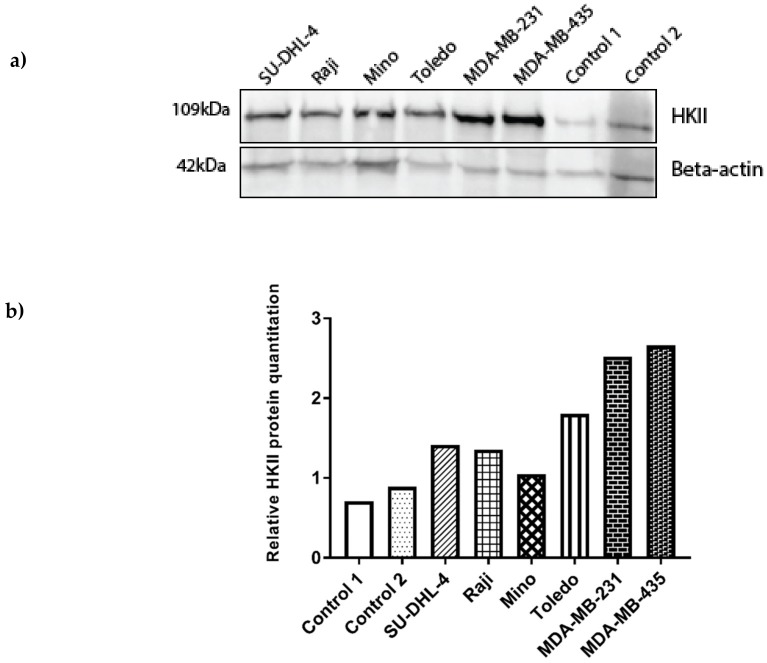
**HKII protein detection.** Western blot analysis of HKII protein (109kDa) expression in NHL (SU-DHL-4, Raji, Mino, Toledo), breast cancer (MDA-MB-231) and melanoma (MDA-MB-435) cell lines compared to healthy control subjects 1 and 2. 30 µg total protein was loaded and samples were normalised against the Beta-actin (42kDa) loading control. (**a**) The HKII protein levels were increased in all four NHL cell lines compared to healthy control subjects 1 and 2, and decreased when compared to the breast cancer (MDA-MB-231) and melanoma (MDA-MB-435) cells. (**b**) HKII protein quantitation analysis compared the normalised intensity ratios between samples and showed the breast cancer (MDA-MB-231) and melanoma (MDA-MB-435) cells to have increased HKII levels as compared to NHL cells and healthy control subjects 1 and 2.

**Figure 4 genes-10-00185-f004:**
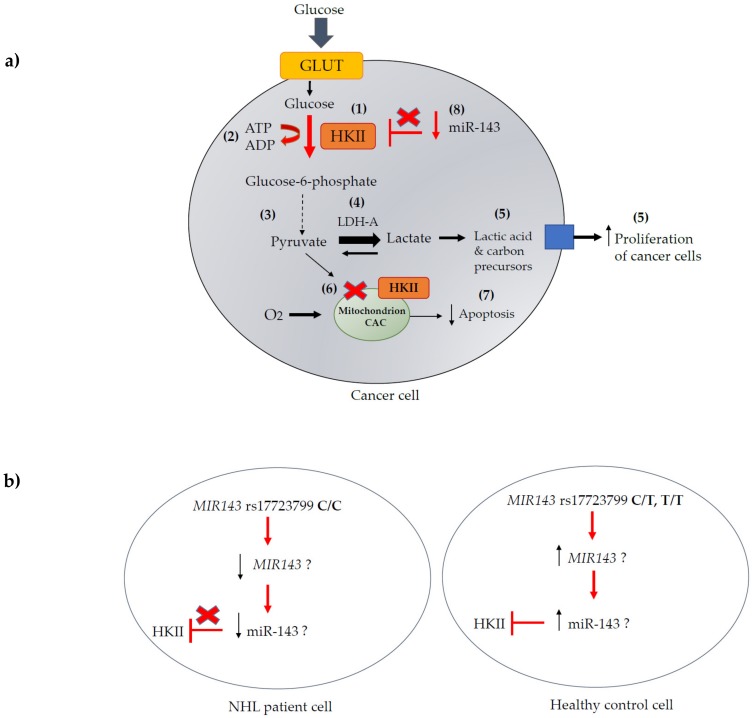
**The role of HKII and miR-143 in cancer.** (**a**) In a cancer cell, aerobic glycolysis occurs with an increased uptake of glucose and production of lactate even in the presence of oxygen (in normal cells glycolysis is anaerobic) and is known as the “Warburg effect”. Upregulation of Hexokinase 2 (*HKII*) in malignant tumours catalyses the conversion of glucose to glucose-6-phosphate (G-6-P) in the first step of the glycolysis pathway (**1**) with high consumption of ATP to ADP (**2**) to maintain the high rate of glycolysis required. G-6-P undergoes multiple conversions to pyruvate (**3**), which together with high lactate dehydrogenase A (LDH-A) activity in one direction (**4**) results in lactic acid production in the cytosol, incorporation of carbon precursors and increased cancer cell proliferation (**5**). Oxidation of pyruvate in the mitochondrion is reduced resulting in reduced citrate production for the citric acid cycle (CAC) (**6**). HKII bound to the mitochondrial membrane can also enhance cancer by increasing mitochondrial resistance to cell death signals helping to reduce apoptosis (**7**). Downregulation of miR-143 has been shown to directly reduce inhibition of *HKII* (**8**) causing increased glycolysis and tumour proliferation. (**b**) In a lymphoid cell, we propose that in the presence of the protective *MIR143* SNP rs17723799 genotype CT or TT, there may be increased *MIR143* expression and miR-143 transcription with increased targeting of *HKII* and reduced glycolysis via the Warburg effect causing decreased lymphomagenesis. This hypothesis is speculative, and further functional studies are required. ATP, adenosine triphosphate; ADP, adenosine diphosphate; CAC, citric acid cycle; GLUT, glucose transporter; LDH-A, lactate dehydrogenase A. (Adapted from Wikimedia [52] and Akins et al. (2018) [23]) [43].

**Table 1 genes-10-00185-t001:** Retrospective Genomics Lymphoma Project-non-Hodgkin lymphoma (GLP-NHL) cohort comprising of different NHL subtypes and the number of cases for each subtype.

NHL Subtype	No. of Cases in the Cohort
FL	95
DLBCL	88
Other B-cell/NHL/unclassified	79
B-CLL	16
T-cell lymphoma	7
Mantle cell lymphoma (MCL)	6
Splenic marginal zone lymphoma (SMZL)	4
Mucosa-associated lymphoid tumour (MALT)	3
Burkitt’s lymphoma (BL)	2
Total	300

GLP: Genomics lymphoma population; NHL: Non-Hodgkin lymphoma; FL: Follicular lymphoma; DLBCL: Diffuse large B cell lymphoma; CLL: Chronic lymphocytic lymphoma.

**Table 2 genes-10-00185-t002:** Sanger sequencing primers for three analysed *MIR143* single nucleotide polymorphisms.

SNP	Forward Primer (5′–3′)	Reverse Primer (5′–3′)	Accession ID
rs3733846	TGTTTGCCTCCATCTCCTCT	CCTTCCCATGGAGCTTTGT	NC_000005.1
rs41291957	CAGGAAACACAGTTGTGAGG ^1^	AGGAGAAGGGGTGTTAGAGG ^1^	NC_000005.1
rs17723799	TGGTCATCCAATCAGCCACC	GGAAGGGACCCTGTCAACTG	NC_000005.1

^1^ Same sequence as MassARRAY^®^ primer design (AgenaCx); SNP: Single Nucleotide Polymorphism.

**Table 3 genes-10-00185-t003:** Basic association testing for 39 genotyped SNPs showing minor allele frequencies (MAFs), Hardy-Weinberg Equilibrium (HWE) score for controls, unadjusted odds ratios, and *p*-values and adjusted *p*-values after correction for multiple testing. A1, minor allele; A2, major allele; MAF, minor allele frequency; NCHROBS, number of chromosomes observed; 1000G, 1000 Genomes Database; HWE, Hardy-Weinberg Equilibrium; OR, odds ratio.

Chr	miRNA/Target Gene	SNP	A1	A2	MAF/NCHROBS	MAF 1000G	HWEUNAFF	OR	*p*-Value	Adjusted *p*-Value
1	*E2F2*	rs2075993	G	A	0.5/862	G = 0.3488/1747	0.2368	0.885	0.4535	1
1	*GEMIN3* 3′-UTR	rs197412	C	T	0.4092/870	C = 0.4744/2376	0.6013	0.8779	0.4301	1
2	hsa-miR-155-3p	rs4672612	A	G	0.338/858	A = 0.3878/1942	0.5476	1.095	0.6006	1
4	*TET2*	rs7670522	A*	C	0.4701/870	C = 0.3600/1803	1.000	1.089	0.6041	1
4	hsa-miR-4330/5100	rs2647257	T	A	0.408/848	T = 0.2386/1195	0.3732	0.9403	0.7094	1
5	hsa-miR-224-5p	rs12719481	G	A	0.2719/868	G = 0.3670/1838	1.000	1.072	0.7052	1
5	hsa-miR-143	rs3733846	G	A	0.1367/878	G = 0.2063/1033	1.000	0.5646	0.012	0.467
5	hsa-miR-143	rs17723799	T	C	0.1129/868	T = 0.1118/560	1.000	0.423	0.0004	0.015
5	hsa-miR-143	rs41291957	A	G	0.1412/878	A = 0.1214/608	1.000	0.5624	0.008	0.326
5	hsa-miR-145	rs353291	C	T	0.4237/826	C = 0.3608/1807	0.1473	1.204	0.2648	1
5	hsa-miR-146-a	rs2910164	C	G	0.2204/862	C = 0.2797/2881	0.5978	1.19	0.3908	1
5	hsa-miR-218-2	rs11134527	A	G	0.2189/868	A = 0.3462/1734	0.7982	1.061	0.7635	1
6	*XPO5*	rs11077	G	T	0.4255/872	G = 0.4036/2021	0.3862	0.9566	0.7869	1
6	*TAB2*	rs9485372	A	G	0.1965/850	A = 0.2408/1206	1.000	0.844	0.4084	1
6	*ESR1, C6orf97*	rs2046210	A	G	0.3353/850	A = 0.4121/2064	1.000	1.277	0.1659	1
8	*TP53*	rs896849	G	A	0.1501/866	G = 0.2183/1093	0.4915	0.9369	0.7695	1
8	*CASC21*	rs13281615	G	A	0.4417/840	G = 0.4912/2460	0.3853	0.7541	0.08388	1
8	*AGO2*	rs3864659	C	A	0.1023/860	C = 0.1436/719	0.6026	1.147	0.6275	1
8	*AGO2*	rs4961280	A	C	0.1835/872	A = 0.1490/746	0.7390	1.416	0.1075	1
9	hsa-miR-101-2	rs462480	G	T	0.3984/876	G = 0.4451/2229	0.7246	1.001	0.9948	1
10	hsa-miR-608	rs4919510	G	C	0.1979/874	G = 0.3638/1822	1.000	1.115	0.6034	1
10	hsa-miR-202	rs12355840	C	T	0.1368/848	C = 0.3189/1597	0.6598	0.494	0.1049	1
11	hsa-miR-210	rs1062099	C	G	0.1701/876	C = 0.1649/826	0.3067	1.29	0.2536	1
11	*LSP1*	rs3817198	C	T	0.3289/836	C = 0.2155/1079	0.7152	0.7085	0.04181	1
11	*TMEM45, BARX2*	rs7107217	A	C	0.4883/858	A = 0.4876/2442	0.5987	0.8992	0.514	1
12	*KRAS 3′-UTR*	rs61764370	C	A	0.0962/852	C = 0.0347/174	1.000	0.6795	0.1473	1
12	hsa-miR-196-a2	rs11614913	T	C	0.4205/880	T = 0.333/1666	0.226	0.928	0.6501	1
12	pre-miR-618	rs2682818	A	C	0.1465/874	A = 0.2424/1214	0.3061	1.115	0.6448	1
14	*HIF1A 3′-UTR*	rs2057482	T	C	0.1250/856	T = 0.2424/1214	0.6797	1.106	0.6831	1
14	*DICER1*	rs3742330	G	A	0.0878/854	G = 0.1382/692	0.2499	1.261	0.4423	1
14	*DICER1*	rs1057035	C	T	0.3709/852	C = 0.1723/863	0.854	0.8726	0.4174	1
16	*TOX3*	rs8051542	T	C	0.4322/856	T = 0.3133/1569	1.000	0.7698	0.1095	1
16	*TOX3*	rs3803662	A	G	0.2207/852	A = 0.4403/2205	1.000	0.9065	0.6119	1
18	hsa-miR-143-5p	rs4987859	T	C	0.0631/856	T = 0.0477/239	0.597	0.744	0.3466	1
18	hsa-miR-27-a-5p	rs4987852	C	T	0.0667/854	C = 0.0190/95	0.4495	1.081	0.811	1
18	hsa-miR-27-a-5p	rs1016860	T	C	0.1175/868	T = 0.1166/584	1.000	0.8423	0.4808	1
21	hsa-miR-155 HG	rs987195	G	C	0.0917/840	G = 0.1472/737	0.361	0.6702	0.1265	1
21	hsa-miR-155	rs12482371	C	T	0.1632/858	C = 0.4151/2079	0.5662	0.7506	0.1748	1
X	hsa-miR-221/222	rs34678647	T	G	0.0375/667	T = 0.1423/537	0.1782	0.433	0.05456	1

**Table 4 genes-10-00185-t004:** Logistic regression analysis of *MIR143* SNPs in association with NHL showing *p*-values after adjusting for covariate ‘sex’. A1, minor allele.

Chr	Gene	SNP	A1	Model	OR (CI 95%)	*p*-Value
5	*MIR143*	rs3733846	G	Additive	0.54 (0.34–0.87)	**0.010**
5	*MIR143*	rs41291957	A	Additive	0.61 (0.39–0.94)	**0.024**
5	*MIR143*	rs17723799	T	Additive	0.61 (0.26–0.71)	**0.0009**

**Table 5 genes-10-00185-t005:** Allele and genotype frequencies of SNPs located in *MIR143* in the GLP-NHL case and control populations in comparison to allele frequencies obtained from HapMap 1000G and gnomAD databases. HWE, Hardy-Weinberg Equilibrium (HWE) test score.

	rs17723799							
	*Allele*			*Genotype*				
	C (%)	T (%)	*p*-Value	C/C (%)	C/T (%)	T/T (%)	*p*-Value	HWE
***Controls***	234 (84.8)	42 (15.2)	**0.013**	99 (71.7) 244 (82.4)	36 (26.1) 48 (16.2)	3 (2.2) 4 (1.4)	**0.039**	1.000 0.311
***Cases***	536 (90.5)	56 (9.5)						
***MAF***	770 (88.7)	98 (11.3)						
***1000G (%)***	88.8	11.2						
***gnomAD (%)***	86.9	13.1						
	**rs3733846**							
	***Allele***			***Genotype***				
	**A (%)**	**G (%)**	***p*-Value**	**A/A (%)**	**A/G (%)**	**G/G (%)**	***p*-Value**	**HWE**
***Controls***	231 (83)	47 (17)	0.057	96 (69) 232 (77)	39 (28) 63 (21)	4 (3) 5 (2)	0.167	1.000 0.785
***Cases***	527 (88)	73 (12)						
***MAF (%)***	758 (86.3)	120 (13.7)						
***1000G (%)***	79.4	20.6						
***gnomAD (%)***	84.4	15.6						
	**rs41291957**							
	***Allele***			***Genotype***				
	**G (%)**	**A (%)**	***p*-Value**	**G/G (%)**	**G/A (%)**	**A/A (%)**	***p*-Value**	**HWE**
***Controls***	229 (82.4)	49 (17.6)	**0.043**	94 (67.6) 233 (77.7)	41 (29.5) 59 (19.7)	4 (2.9) 8 (2.6)	0.070	1.000 0.106
***Cases***	525 (87.5)	75 (12.5)						
***MAF (%)***	754 (85.9)	124 (14.1)						
***1000G (%)***	87.9	12.1						
***gnomAD (%)***	84	16						

**Table 6 genes-10-00185-t006:** Mixed model logistic regression analysis adjusted for the covariate ‘sex’ for the rs17723799 SNP in *MIR143*.

Model	Genotype	Controls (%)	Cases (%)	χ^2^	OR (95% CI)	*p*-Value
**Allelic**	T vs. C	39/459	36/178	12.84	-	**0.0003**
**Additive**	-	-	-		0.43 (0.26–0.71)	**0.0009**
**Co-dominant**	CC	97 (71.9)	243 (82.4)		1.00 (reference)	
	CT	35 (25.9)	48 (16.3)		0.55 (0.33–0.91)	
	TT	3 (2.2)	4 (1.4)		0.47 (0.10–2.23)	0.051
**Dominant**	CC	97 (71.9)	243 (82.4)		1.00 (reference)	
	CT–TT	38 (28.1)	52 (17.6)		0.54 (0.33–0.88)	**0.015**
**Recessive**	CC–CT	132 (97.8)	291 (98.6)		1.00 (reference)	
	TT	3 (2.2)	4 (1.4)		0.53 (0.11–2.52)	0.438
**Over-dominant**	CC–TT	100 (74.1)	247 (83.7)		1.00 (reference)	
	CT	35 (25.9)	48 (16.3)		0.56 (0.33–0.92)	**0.024**
**Log-additive**	-	135 (31.4)	295 (68.6)		0.58 (0.38–0.90)	**0.017**

**Table 7 genes-10-00185-t007:** Linkage disequilibrium (D’) values for SNPs on chromosome 5 in HapMap 1000G European (EUR) Central European (CEU)/Great Britain (GBR) populations.

SNP	Location (GRChg38)	rs3733846	rs12659504	rs878008	rs17723799	rs41291957
***rs3733846***	149,425,059	-	1.000/1.000	1.000/1.000	1.000/1.000	0.721/0.941
***rs12659504***	149,425,442	-	-	1.000/1.000	1.000/1.000	0.721/0.942
***rs878008***	149,425,488	-	-	-	1.000/1.000	0.721/0.942
***rs17723799***	149,427,514	-	-	-	-	0.999/0.999
***rs41291957***	149,428,827	-	-	-	-	-

**Table 8 genes-10-00185-t008:** Summary statistics for three European NHL-GWAS with an imputation scan (InterLymph Consortium) and meta-analysis (Cerhan et al. 2014) for two SNPs in high LD with rs17723799 on chromosome 5. EAF, effect allele frequency; OR, odds ratio, CI, confidence interval.

SNP	Chr	Location (GRChg38)	Group	Controls	Cases	Effect Allele	EAF Controls	EAF Cases	OR	CI (95%)	*p*-Value
***rs12659504***	5	149,425,442	NCI	6221	2661	G	0.1502	0.1359	0.93	0.84–1.02	0.120
***rs12659504***	5	149,425,442	GELA	525	548	G	0.1418	0.125	0.90	0.70–1.15	0.392
***rs12659504***	5	149,425,442	MAYO_DLBCL	171	392	G	0.173	0.1569	0.79	0.54–1.14	0.211
***rs12659504***	5	149,425,442	SF	747	254	G	0.133	0.1205	0.89	0.66–1.21	0.456
***rs12659504***	5	149,425,442	Meta-analysis	7664	3855				**0.91**	**0.84–0.99**	**0.033**
***rs878008***	5	149,425,488	NCI	6221	2661	C	0.1501	0.1362	0.93	0.84–1.02	0.132
***rs878008***	5	149,425,488	GELA	524	548	C	0.1396	0.1242	0.91	0.71–1.16	0.445
***rs878008***	5	149,425,488	MAYO_DLBCL	172	392	C	0.172	0.1548	0.78	0.53–1.13	0.188
***rs878008***	5	149,425,488	SF	747	253	C	0.1307	0.1199	0.90	0.66–1.23	0.516
***rs878008***	5	149,425,488	Meta-analysis	7664	3854				**0.92**	**0.84–1.00**	**0.041**

**Table 9 genes-10-00185-t009:** Haplotype frequencies for analysed *MIR143* SNPs in GLP-NHL case and control populations.

Haplotype	rs3733846	rs41291957	rs17723799	Frequencies Cases	Frequencies Controls
***1***	A	A	C	0.03608	0.03255
***2***	A	A	T	0.00000	0.00000
***3***	A	G	C	0.81643	0.79232
***4***	A	G	T	0.02582	0.00517
***5***	G	A	C	0.02159	-
***6***	G	A	T	0.06733	0.14260
***7***	G	G	C	0.03092	0.02067
***8***	G	G	T	0.00182	0.00670

**Table 10 genes-10-00185-t010:** Association of haplotypes for analysed *MIR143* SNPs in GLP-NHL case and control populations.

Haplotype	rs3733846	rs41291957	rs17723799	Haplotype Frequencies	OR (CI 95%)	*p*-Value
***1***	A	G	C	0.80883	0.90 (0.43–1.90)	0.7799
***2***	A	G	T	0.01898	6.34 (0.60–67.07)	0.1246
***3***	G	A	C	0.01461	Inf (Inf-Inf)	0.0000
***4***	**G**	**A**	**T**	**0.09137**	**0.42 (0.18–1.00)**	**0.0495**
***5***	G	G	C	0.02762	1.22 (0.38–3.88)	0.7370
***rare***	*	*	*	0.00359	0.12 (0.00–2.95)	0.1920

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
