# Peer review of "Single Nucleotide Polymorphisms in MIR143 Contribute to Protection against Non-Hodgkin Lymphoma (NHL) in Caucasian Populations"

_genes, 2019, doi:10.3390/genes10030185_

Round 1
Reviewer 1 Report
The revised version of this manuscript authors satisfactorily response all the quarry raised by the reviewer except two comments (from reviewer no.2 comment no 7 and 8). Reviewer understand the author limitation. But without have done these experiment reviewer thinks that the conclusion is not strong enough with the current data.
Author Response
The revised version of this manuscript authors satisfactorily response all the quarry raised by the reviewer except two comments (from reviewer no.2 comment no 7 and 8). Reviewer understand the author limitation. But without have done these experiment reviewer thinks that the conclusion is not strong enough with the current data.
We appreciate the opportunity to further clarify this important point for the Reviewer. As we outlined in our previous responses to Reviewer 2’s comments no. 7 and 8: Question 7 suggested we provide NHL subtype examination of miR-143 expression from the literature or as an alternative, in the sample cohort. Question 8, expanded on this and sought to have gene expression measured in patient and control lymphocytes. We address these in general below, and in more detail in response to the specific questions.
In response to Question 7, there is an abundance of literature describing the tumour suppressive role of miR-143 through multiple targets in other cancers and pathologies, but not in NHL. In light of this we have now included the following relevant data in the revised version of the manuscript in the Discussion (Page 16, Line 503):
“Although not in NHL but as a marker of non-small cell lung cancer (NSCLC), miR-143 expression in peripheral blood mononuclear cells (lymphocytes and monocytes) was significantly lower in NSCLC patients than in healthy individuals (p < 0.0001) [44]. Similarly a study in renal cell carcinoma (RCC) identified the miR-143/145 cluster to be downregulated in RCC tissues compared to adjacent non-cancerous tissues, with significantly higher HKII levels in RCC tissues compared to non-cancerous tissues, confirming the tumour suppressive effect of the miR-143/145 cluster through targeting HKII [45]. One study in aggressive Burkitt’s lymphoma (BL) showed that upregulated miR-143 by regulation of PI3K/Akt prevented cell growth, implicating miR-143 as a tumour suppressor in NHL [46].”
In response to Question 8, we agree with the Reviewer that we cannot conclude a direct functional association between miR-143 levels and target HKII expression with lymphoma onset in patients, nor can we conclude the rs17723799 genotypes are associated with phenotype in NHL cell lines. In addition we had revised and updated our Figure (Figure 4) in the Discussion to suggest how regulatory mechanisms might occur based on the literature in other cancer types. We have now revised the Conclusion to state more clearly our specific findings and conclusions (Page 17, Line 565)
“This study is the first to report significant statistical correlation between SNPs in MIR143 and reduced risk of NHL in Caucasians, and is supported by identification of significant SNPs in high LD in a large European NHL GWAS meta-analysis study. Our findings suggest that the three SNPs in LD in MIR143 may be novel useful biomarkers to assess risk of NHL in a clinical setting. Furthermore, this study gives some insight into MIR143 transcriptional regulation by a protective promoter polymorphism in NHL. We do not suggest that there is a definitive functional association between MIR143 SNPs and lowered NHL onset, however further functional analyses are necessary to confirm this potential mechanism.”
Reviewer 2 Report
This paper examines the association between polymorphisms previously implicated in tumorigenesis and the risk of lymphoma in a heterogeneous series including different non-Hodgkin lymphoma subtypes in comparison to healthy controls. The results are potentially interesting. Given for good the association of the three SNPs with miR-143 level, it has to be better framed how a different level of miR-143 and miR-145 impact on lymphoma onset. 1- The data show that three sequence variants close to miR-143 and the low level of miR-143 (and miR-145?) confer protection to lymphoma onset. Is miR-143 (and miR-145) differentially expressed in subtypes of NHLs according to literature data? Comment this aspect. 2- miR-143 and miR-145 are embedded in the long noncoding RNAs CARMN and MIR143HG (Ounzain S et al. CARMEN, a human super enhancer-associated long noncoding RNA controlling cardiac specification, differentiation and homeostasis, J. Mol. Cell. Cardiol, 2015; and Rani N et al. A primate lncRNA mediates notch signaling during neuronal development by sequestering miRNA, Neuron 2016). Can you recognize an implication of miR-143 in the Notch pathway in lymphoma? 3- If the three SNPs are responsible of a lower level of miR-143 and mR-145 , which are DNA-binding proteins and transcription factors potentially involved? 4- Lymphoma originate from different lymphocyte types and lymphocytes at different maturation stages. Provide details about the association of the three mir-143 polymorphisms with NHL subtypes. 5- In Figure S2, it is shown the expression level of MIR143HG as reported in GTEx. MIR143HG is not equivalent to the microRNA miR-143. For example, miR-143 and miR-145 were found to have opposite expression patterns from the lncRNA during cardiogenesis. The relationship between miR-143/miR-145 and miR143HG is an open issue. This means that citations of the Figure S2 in the text and caption of Figure S2 are not correct. Similarly, evaluate if Figure S1 is correct.Author Response
This paper examines the association between polymorphisms previously implicated in tumorigenesis and the risk of lymphoma in a heterogeneous series including different non-Hodgkin lymphoma subtypes in comparison to healthy controls. The results are potentially interesting. Given for good the association of the three SNPs with miR-143 level, it has to be better framed how a different level of miR-143 and miR-145 impact on lymphoma onset.
1- The data show that three sequence variants close to miR-143 and the low level of miR-143 (and miR-145?) confer protection to lymphoma onset. Is miR-143 (and miR-145) differentially expressed in subtypes of NHLs according to literature data? Comment this aspect.
We would like to thank the Reviewer for this comment referring to whether miR-143 and miR-145 are differentially expressed in different NHL subtypes. We would like to confirm that our study suggests potential regulation of miR-143 by promoter polymorphisms to increasemiR-143 and thereby confer protection against NHL. At present there is no data available on miR-143/145 expression in NHL and its main subtypes, however there is an abundance of literature describing the tumour suppressive role of miR-143 through multiple targets in other cancers and pathologies as outlined above in response to Reviewer 1, and we have now outlined one available study on Burkitt’s lymphoma in the Discussion (Page 16, Line 509):
“One study in aggressive Burkitt’s lymphoma (BL) showed that upregulated miR-143 by regulation of PI3K/Akt prevented cell growth, implicating miR-143 as a tumour suppressor in NHL [46].”
2- miR-143 and miR-145 are embedded in the long noncoding RNAs CARMN and MIR143HG (Ounzain S et al. CARMEN, a human super enhancer-associated long noncoding RNA controlling cardiac specification, differentiation and homeostasis, J. Mol. Cell. Cardiol, 2015; and Rani N et al. A primate lncRNA mediates notch signaling during neuronal development by sequestering miRNA, Neuron 2016). Can you recognize an implication of miR-143 in the Notch pathway in lymphoma?
We would like to thank the Reviewer for providing this information and referring to other studies that may be of interest. From what we can find in the literature, NOTCH pathway mutations have very recently been identified in mature B-cell malignancies through the application of next-generation sequencing [Arruga, F. The Notch Pathway and Its Mutations in Mature B-cell Malignancies. 2018. Frontiers in Oncology]. We agree it may be potentially interesting to determine whether miR-143/-145 targets genes in this pathway.
3- If the three SNPs are responsible of a lower level of miR-143 and mR-145, which are DNA-binding proteins and transcription factors potentially involved?
In response to this we have now included more detail on miR-143/145 transcription factors in the Discussion as per below (Page 15, Line 472):
“In addition, transcriptional regulation of miR-145 was recently reviewed by Zeinali et al. [43]. The authors discuss miR-145 regulation by DNA-binding factors including c-MYC, p53, forkhead transcription factors of the O class 1 and 3 (Fox01 and Fox03) and miR-143/145 level regulation by RREB1 in other cancers but not in lymphoma.”
4- Lymphoma originate from different lymphocyte types and lymphocytes at different maturation stages. Provide details about the association of the three mir-143 polymorphisms with NHL subtypes.
In response to this we have now undertaken common subtype analysis for the three MIR143polymorphisms in association with DLBCL and FL risk. Analysis for all SNPs showed non-significant association (p > 0.05) which could have been due to the reduced power due to the small sample size of DLBCL and FL patients in the case cohort. This information has now been included in the Results as per below (Page 7, Line 286):
“Analysis of the three SNPs was also performed in association with the two most common NHL subtypes present in the cohort, i.e. DLBCL and FL, however results were non-significant for all three SNPs (p > 0.05). This could be due to the small number of cases representing these subtypes and the reduced statistical power.”
5- In Figure S2, it is shown the expression level of MIR143HG as reported in GTEx. MIR143HG is not equivalent to the microRNA miR-143. For example, miR-143 and miR-145 were found to have opposite expression patterns from the lncRNA during cardiogenesis. The relationship between miR-143/miR-145 and miR143HG is an open issue. This means that citations of the Figure S2 in the text and caption of Figure S2 are not correct. Similarly, evaluate if Figure S1 is correct.
We would like to thank the Reviewer for their comment regarding the appropriateness of the supplementary figures in this manuscript. We agree that it would be incorrect to claim that MIR143host gene expression would directly relate to levels of mature miR-143/-145 expression in whole blood and EBV-transformed lymphocytes, and therefore Figure S2 has been removed from the manuscript. Similarly, the increased expression data shown in the GTEx portal for the 3 SNP eQTLs are for the MIR143HG and not mature miR-143/-145 expression (Figure S1). We have now revised the Discussion to clarify the relationship between miR-143/-145 and its MIR143HG (Page 16, Line 520):
“Although the relationship between miR-143/145 and its MIR143 host gene is an open issue, the potentially increased MIR143 host gene expression as seen due to the polymorphisms in skeletal muscle may enhance hsa-miR-143-3p and -5p processing, which poses a potential protective function for these SNPs in lymphocytes. Presence of the rs17723799 C/T or T/T genotype may increase MIR143 expression and miR-143 transcription with subsequent downregulation of HKII causing reduced glycolysis and lymphomagenesis (Figure 4b).”
Also the caption for Figure S1 has been updated accordingly (Page 18, Line 578):
Figure S1. Significant eQTLs for 3 analysed MIR143SNPs. a) rs17723799, b) rs3733846 and c) rs41291957 in 491 subjects showing increased MIR143hostgene expression with heterozygous (Het) and homozygous alternate (Homo Alt) genotypes in skeletal muscle (GTEx Portal).
Additional references now included in the manuscript:
43. Zeinali, T.; Mansoori, B.; Mohammadi, A.; Baradaran, B. Regulatory mechanisms of miR-145 expression and the importance of its function in cancer metastasis. Biomedicine & Pharmacotherapy 2019, 109, 195-207, doi:https://doi.org/10.1016/j.biopha.2018.10.037.
44. ZENG, X.-l.; ZHANG, S.-y.; ZHENG, J.-f.; YUAN, H.; WANG, Y. Altered miR-143 and miR-150 expressions in peripheral blood mononuclear cells for diagnosis of non-small cell lung cancer. Chinese Medical Journal 2013, 126, 4510-4516, doi:10.3760/cma.j.issn.0366-6999.20122931.
45. Yoshino, H.; Enokida, H.; Itesako, T.; Kojima, S.; Kinoshita, T.; Tatarano, S.; Chiyomaru, T.; Nakagawa, M.; Seki, N. Tumor-suppressive microRNA-143/145 cluster targets hexokinase-2 in renal cell carcinoma. Cancer Science 2013, 104, 1567-1574, doi:doi:10.1111/cas.12280.
46. dos Santos Ferreira, A.C.; Robaina, M.C.; de Rezende, L.M.M.; Severino, P.; Klumb, C.E. Histone deacetylase inhibitor prevents cell growth in Burkitt’s lymphoma by regulating PI3K/Akt pathways and leads to upregulation of miR-143, miR-145, and miR-101. Annals of Hematology 2014, 93, 983-993, doi:10.1007/s00277-014-2021-4.
This manuscript is a resubmission of an earlier submission. The following is a list of the peer review reports and author responses from that submission.
Round 1
Reviewer 1 Report
The authors have investigated a panel of 39 SNP previously related to miRNAs in cancers. Three of these SNPs related with miR-143 were associated to reduced NHL susceptibility. Since the enzyme HKII is upregulated in many cancer types and a potential target of miR-143, the expression of the enzyme was assessed in cancer cell lines with different combination of alleles. The Discussion of the results should be reconsidered removing the excess of speculative reasonings and more focused on the significant SNPs, miR-143 and its relevance in the NHL.
The authors should consider the following points:
1- The scheme in Figure 1 does not include HKII and its functional role.
2- NHLs subtypes are characterized by different cell of origin, driver genetic abnormalities, gene expression patterns and prognosis. So the frequencies of the NHL subtypes present in the NHL cohorts may be decisive in association studies. Summarize in a Table the number of cases belonging to different NHL subtypes in your cohort. Describe how are the eQTL associated to miRNAs in eachNHL subtypes. Compare your cohort and that of other studies reporting significant association between miRNAs SNP and NHL.
3- You have cited Li T et al. (Ref 18) since this study reported that the SNP rs11614913, located in the coding sequence of miR-196a2, is a marker of susceptibility in NHLs. That study compared about 300 NHL and 300 healthy individuals, numbers similar to that of the present study. How do you explain the lack of association of the SNP rs11614913 in your study (p value=0.65)? This SNP could be considered a positive control of association with NHL for your study. Comment this point.
4- The SNP rs11614913 of miR-196a2 modify the mature form miR-196a-3p (previously miR-196a* according to miRBase). This sequence variation probably modify the expression/maturation of the miRNA and the spectrum of genes targeted by miR-196a-3p or abolish its biological activity. The SNP rs17723799 is a variant distant thousands of bases from miR-143 that could affect the transcription regulation of the miRNA. This is a big difference to underline.
5- You compare the expression level of HKII in 4 NHL cell lines, 1 melanoma cell line, 1 breast cancer cell line and 2 lymphocytes samples from healthy individuals. The expression is higher in the breast cancer cell line homozygous for the SNP rs17723799 than the other samples. At protein level, the pattern observed in the mRNA analysys is partially confirmed. To establish a relationship between miR-143 and its potential target HKII, it you should be measure the level of miR-143-5p and miR-143-3p in cell lines and reference samples. Without these data, the claims present in Discussion about these results seems very speculative.
6- A more appropriate reference for the breast cancer cell line homozygous for rs17723799 are breast cancer cell lines homozygous for the major allele. Differences of HKII expression may regard the tissue of origin of the cell lines independently from the gene targeting by miR-143.
7- The SNP rs17723799 is about one thousand bases far from miR-143 but this single substitution could affect its transcription. If miR-143 level was expressed at levels lower than normal in NHLs, miR-143 loss would be responsible of increased level of HKII. Is miR-143 downregulated in the NHL subtypes? This information can be obtained from literature data. Alternatively, you should assess the miR-143 expression in your cohort or in other samples representative of the NHL.
8- Page 15, line 369, you have written that the SNP rs17723799 overlaps with a transcription factor binding site. Explain which transcription factor you refer to
9- In Discussion, you provide data about miR-143 in different cancer types but not in the subtypes of NHL that is the subject of this study. How is the expression level of mIR-143 in NHL? The discussion is should be focused on the SNPs associated to NHL, miR-143 and HKII in NHL subtypes. In particular, the paragraph starting at line 400, page 15, should be reconsidered.
10- You do not comment about the association of the SNP rs4987852 with miR-143. The SNP rs4987852 is not significantly associated to the NHL, however this could depends from the heterogeneity in the cohort. This genetic variant is within the 3'-UTR of the gene BCL2, a protein of relevance for NHL in general and tipically overexpressed in follicular lymphoma. To note that BCL2 is targeted by miR-143-3p.
11- When you are describing the gene, miR-143 is sufficient, but when you describe the expression you should specify to which mature form, miR-143-5p and miR-143-3p, you refer.
Reviewer 2 Report
This article represents a case-control study aiming to evaluate the association of miRNA polymorphisms (miRSNPs) in non-Hodgkin lymphoma (NHL) in Australian Caucasian population. Although this association has been studied especially during the last decade, the results have been interesting with new three miRSNPs in the miR-143. However, this reviewer has several major concerns that authors should address further.
Major comment:
1) In the abstract author described in the line 18-21 about the odd ratio of all three miRSNPs in univariate analysis but the data is not presented in in the manuscript or in the table. Author said also the rs17723799, remained significant after correction for multiple testing (p-value = 0.015) but the data is missing in the table. Author are advised to put a multivariate analysis table separately for easy to understand for the reader.
2) Introduction section is too long. Author should shorter the introduction with specific information relevant to this work. (for example, there is no need to explain the miRNA function in details rather explain what is the role of Hexokinase-II in NHL).
3) Author already described the aim of this work in the introduction, but he aims of this work is not specifically pin pointed. Author should be more specifically rewrite the aims for easily understandable to the general reader. (Like, why author discussed LDH-A in this aim section)
4) Figure 1 is not necessary to put in this manuscript. This figure explains the general Warburg effect. Author should remove this figure during the revision.
5) Reviewer is not clear about the insertion of the MAF 1000G data and the gnomAD data in the table 1. If it’s for comparison of earlier study and current study, then authors advised to discuss the data in the results section.
6) Authors are recommended to cite the tables in ascending numeric order upon first appearance in the manuscript file for understanding the data or the text.
7) SNP, rs17723799 present in the promoter region of the miR143. Author should show the expression levels of the miR143 in the studied NHL patient to conclude the rs17723799 is directly regulating miRNA expression levels in the patient samples.
8) As per reviewer understanding author want to establish a correlation between the rs17723799 (C>T) genotype and their miR143 target gene HKII expression levels. TO show the direct impact in the patient author advised to take lymphocytes from CC and TT genotypes NHL patients and show the HKII expression levels in both type of patient and compare the expression with healthy control lymphocytes.
Minor comments:
9) In line no 88 authors mentioned “88 HKII is an enzyme involved in cancer cell glycolysis and the Warburg effect [23] (Figure 1” But in the figure there is no HKII.
10) Line 139. “Prior to each run, six primer extension products were initially spotted on to the SpectroCHIP®”. Which six primer extension? What does mean the primer extension product? Is it complete PR product?
11) Authors mentioned in the line 205-207 that comparison of experiment data and observed Hapma 1000G database and genomeAD in table 2. But there is no such data in table 2.
12) There are several misused characters in the manuscript. The authors should carefully revise their manuscript before the submission.